# Neurological Music Therapy Rebuilds Structural Connectome after Traumatic Brain Injury: Secondary Analysis from a Randomized Controlled Trial

**DOI:** 10.3390/jcm11082184

**Published:** 2022-04-14

**Authors:** Aleksi J. Sihvonen, Sini-Tuuli Siponkoski, Noelia Martínez-Molina, Sari Laitinen, Milla Holma, Mirja Ahlfors, Linda Kuusela, Johanna Pekkola, Sanna Koskinen, Teppo Särkämö

**Affiliations:** 1Cognitive Brain Research Unit, Department of Psychology and Logopedics, Faculty of Medicine, University of Helsinki, 00014 Helsinki, Finland; sini-tuuli.siponkoski@helsinki.fi (S.-T.S.); noelia.martinez@upf.edu (N.M.-M.); teppo.sarkamo@helsinki.fi (T.S.); 2Centre of Excellence in Music, Mind, Body and Brain, University of Jyväskylä & University of Helsinki, 00014 Helsinki, Finland; sari.laitinen@espoo.fi; 3School of Health and Rehabilitation Sciences, Queensland Aphasia Research Centre and UQ Centre for Clinical Research, The University of Queensland, Brisbane, QLD 4029, Australia; 4Espoo Hospital, 02740 Espoo, Finland; 5Independent Researcher, 00550 Helsinki, Finland; milla.holma@elisanet.fi; 6Independent Researcher, 02330 Espoo, Finland; mirja.ahlfors@kolumbus.fi; 7Department of Physics, University of Helsinki, 00014 Helsinki, Finland; linda.kuusela@hus.fi; 8HUS Medical Imaging Center, Department of Radiology, Helsinki Central University Hospital and University of Helsinki, 00014 Helsinki, Finland; pekkojo@gmail.com; 9Clinical Neuropsychology Research Group, Department of Psychology and Logopedics, Faculty of Medicine, University of Helsinki, 00014 Helsinki, Finland; sanna.koskinen@helsinki.fi

**Keywords:** music therapy, traumatic brain injury, TBI, executive function, rehabilitation, structural connectivity, connectometry, DTI

## Abstract

Background: Traumatic brain injury (TBI) is a common and devastating neurological condition, associated often with poor functional outcome and deficits in executive function. Due to the neuropathology of TBI, neuroimaging plays a crucial role in its assessment, and while diffusion MRI has been proposed as a sensitive biomarker, longitudinal studies evaluating treatment-related diffusion MRI changes are scarce. Recent evidence suggests that neurological music therapy can improve executive functions in patients with TBI and that these effects are underpinned by neuroplasticity changes in the brain. However, studies evaluating music therapy induced structural connectome changes in patients with TBI are lacking. Design: Single-blind crossover (AB/BA) randomized controlled trial (NCT01956136). Objective: Here, we report secondary outcomes of the trial and set out to assess the effect of neurological music therapy on structural white matter connectome changes and their association with improved execute function in patients with TBI. Methods: Using an AB/BA design, 25 patients with moderate or severe TBI were randomized to receive a 3-month neurological music therapy intervention either during the first (AB, *n* = 16) or second (BA, *n* = 9) half of a 6-month follow-up period. Neuropsychological testing and diffusion MRI scans were performed at baseline and at the 3-month and 6-month stage. Findings: Compared to the control group, the music therapy group increased quantitative anisotropy (QA) in the right dorsal pathways (arcuate fasciculus, superior longitudinal fasciculus) and in the corpus callosum and the right frontal aslant tract, thalamic radiation and corticostriatal tracts. The mean increased QA in this network of results correlated with improved executive function. Conclusions: This study shows that music therapy can induce structural white matter neuroplasticity in the post-TBI brain that underpins improved executive function.

## 1. Introduction

Traumatic brain injury (TBI) is a common and devastating neurological disorder, affecting over 50 million people each year worldwide [1], with often poor long-term outcomes [2]. The primary neuropathology associated with TBI is structural white matter damage, that is, axonal injury, established already half a century ago by the seminal post-mortem studies [3,4]. In TBI, the white matter damage disrupts effective neural communication and impairs neural networks that link brain structure to function, typically causing deficits in cognitive, social, and emotional functioning [5,6,7]. Among the most common, persistent, and disabling aspects of cognitive impairment following TBI is executive dysfunction [8], which is often caused by diffuse axonal injury (DAI) resulting in widespread connectivity deficits in the brain [9,10].

Due to the neuropathology of TBI, neuroimaging plays a crucial role in its assessment. In the acute setting, MRI scans are used to guide appropriate management by detecting brain injuries that require neurosurgical interventions or further monitoring. However, routinely acquired MRI might not reveal findings even in patients with symptoms due to the DAI mechanism [11,12]. Therefore, advanced neuroimaging techniques reflecting white matter structures such as diffusion tensor imaging (DTI) have been under active research in TBI. Studies have shown that TBI patients have structural connectivity deficits in multiple white matter tracts, most commonly in long coursing and commissural fibres that are most vulnerable to injury in TBI [12,13,14]. After the initial injury, the degeneration of white matter tracts persists for years [15,16] and is associated with poor long-term functional and cognitive outcomes in TBI [16,17].

While DTI has been used to improve the diagnostics and classification system of TBI, very little research has thus far been carried out in determining treatment-induced white matter neuroplasticity changes in TBI. Given the dynamic nature of DAI, intervention studies charting the possible discontinued deterioration or recovery of white matter injury over time would be of great interest. Ultimately, this information would help to target clinical interventions for rehabilitation. A recent animal study suggests that cognitive TBI treatments can induce white matter plasticity [18], but to our best knowledge, studies on treatment-induced structural white matter neuroplasticity in TBI patients have not been published.

Cognitive therapies have emerged as efficient treatments to restore cognitive functions and improve functional outcomes in TBI [19,20]. In cognitive neurological rehabilitation, music has emerged as a viable and applicable tool during the past decades, partly owing to its capacity to engage widespread neural networks across bilateral cortical and subcortical areas [21]. Research findings in stroke patients suggest that music-based interventions engage an array of cognitive functions, resulting in cognitive improvement [22,23] and structural and functional neuroplasticity changes [24,25] in the damaged brain. In our recent randomized controlled trial (RCT) on patients with moderate-to-severe TBI, we found that a 3-month neurological music therapy (NMT) intervention enhanced executive function and increased structural grey matter neuroplasticity in prefrontal areas [26] as well as normalized or enhanced functional connectivity in the brain, especially in frontal and parietal regions [27]. Given the extent of the brain regions and pathways stimulated by music, it is possible that NMT may induce also more widespread structural connectivity changes in TBI, but this has not been studied previously.

Here, using longitudinal diffusion MRI (dMRI) data from our previous RCT [26,27], we set out to determine as a secondary outcome of the trial NMT-induced structural white matter connectivity changes and their association with improved executive function in a sample of 25 TBI patients with a 6-month follow-up. To do this, we carried out white matter connectometry analysis utilizing quantitative anisotropy (QA), which has been shown to be superior to conventional single-tensor based or tract-based analysis [28]. Connectometry analysis utilizes permutation testing to identify group differences in white matter tracts across the whole brain and has been used in neurological patients, for example, to uncover white matter tracts subserving word production [29] and verb retrieval [30] in post-stroke aphasia. Based on our previous findings of increased grey matter volume and functional connectivity, which were most evident in right prefrontal areas, after the NMT [26,27], we hypothesized that it would induce structural connectivity changes especially in the right frontal and dorsal pathways.

## 2. Materials and Methods

### 2.1. Subjects and Study Design

Forty TBI patients from the Helsinki and Uusimaa Region of Finland were recruited through the Brain Injury Clinic of the Helsinki University Central Hospital (HUCH), Validia Rehabilitation Helsinki, and the Department of Neurology of the Lohja Hospital during 2014–2017 to this RCT (NCT01956136). The inclusion criteria were: (1) diagnosed TBI according to the International Statistical Classification of Diseases and Related Health Problems, 10th revision (ICD-10), fulfilling the criteria of at least moderate severity (Glasgow Coma Scale [GCS] score: ≤12 and/or loss of consciousness >30 min and/or post-traumatic amnesia [PTA] ≥24 h and positive findings on CT/MRI); (2) time since injury ≤24 months at the time of recruitment; (3) cognitive symptoms caused by TBI (attention, executive function, memory); (4) no previous neurological or severe psychiatric illnesses or substance abuse; (5) age 16–60 years; (6) native Finnish speaking or bilingual with sufficient communication skills in Finnish; (7) living in the Helsinki-Uusimaa area; and (8) understanding the purpose of the study and being able to give an informed consent. Patients with GCS score 9–12 and/or loss of consciousness 30 min–24 h and/or PTA 1–7 days and abnormal structural imaging on CT/MRI were defined as having a moderate TBI, and patients with GCS score 1–9 and/or loss of consciousness >24 h and/or PTA > 7 days and abnormal structural imaging on CT/MRI were defined as having a severe TBI [31]. Both the extended Glasgow Outcome Scale (GOSE) [32] and the Neurological Outcome Scale for Traumatic Brain Injury (NOS-TBI) [33] were administered to obtain information of the overall symptoms and current functional outcome after TBI. The trial was conducted according to the Declaration of Helsinki and was consistent with good clinical practice and the applicable regulatory requirements. The trial protocol was approved by the Coordinating Ethics Committee of the Hospital District of Helsinki and Uusimaa (reference number 338/13/03/00/2012) and all participants signed an informed consent.

The study was a single-blind crossover RCT with a 6-month follow-up period. During 2014–2017, 4994 patients with TBI were screened for eligibility, 190 met the inclusion criteria, and 40 were randomized to the AB (*n* = 20) and BA (*n* = 20) groups. The randomization was stratified for lesion laterality and performed using a random number generator by a person not involved in patient recruitment or assessments. To ensure steady allocation to both groups across the trial, the randomization was done in batches of two consecutive patients. After the baseline measurements at time point 1 (TP1), which included MRI scans and neuropsychological assessments, the AB group received NMT in addition to standard care for the first 3 months, whereas the BA group received only standard care. At the 3-month crossover point (TP2), follow-up measurements using the same outcome measures were carried out. After this, the BA group received NMT and standard care for 3 months and the AB group received only standard care. At the 6-month completion point (TP3), the measurements were carried out once again. All assessments were carried out by research personnel blinded to the patients’ group allocation. Due to the nature of the intervention, patients were not blinded. Standard care comprised any physical therapy, occupational therapy, speech therapy or neuropsychological rehabilitation which the patients received in public (or private) healthcare during the study period. There were no statistically significant differences between the AB and BA groups in the amount of received standard care [26].

Out of the 40 randomized patients, 1 participant dropped out before the TP1 measurements, 2 participants dropped out before TP2, and another 3 participants dropped out before TP3. The dropouts were mainly due to lack of energy and motivation. All dropouts (*n* = 6) occurred in the BA group, which was likely linked to the long waiting period before the intervention. Of these, five took place before the onset of the intervention. Of the remaining 34 patients, 1 was excluded from the analyses due to intensive self-implemented piano training, which was not part of the trial protocol, and 8 were excluded from the analyses due to lack of MRI data owing to contraindications or technical difficulties during the scanning. Finally, 25 patients (AB: *n* = 16, BA: *n* = 9) completed the MRI acquisition in the three time points and were included in the present study. The flowchart of the included patients with TBI is shown in Figure 1. 

### 2.2. Intervention

The NMT intervention is described in detail in our previous publication [26]. Briefly, it consisted of 20 individual therapy sessions (2 times/week, 60 min/session) held by a trained music therapist at Validia Rehabilitation Helsinki. No previous musical experience was required to participate in the NMT. The focus of the NMT was on active musical production using different instruments (drums, piano) in three training modules involving (i) rhythmical training (playing sequences of musical rhythms and coordinated bimanual movements on a djembe drum and on own body), (ii) structured cognitive-motor training (playing musical exercises on a drum set with varying levels of movement elements and composition of drum pads), and (iii) assisted music playing (learning to play own favourite songs on the piano with the help of figure notes). All modules also included musical improvisation to facilitate more creative and interactive musical expression The difficulty level of the exercises was initially adjusted and then increased in a stepwise manner within and across the NMT sessions, to meet the skill level and progression of the patient.

### 2.3. Neuropsychological Assessment

The primary behavioural outcome measure was change in performance on the Frontal Assessment Battery (FAB) [34]. Assessing global executive function and being applicable across all severity levels of TBI, the FAB measures different aspects of frontal lobe functions and consists of six subtests exploring conceptualization (similarities subtest), mental flexibility (lexical fluency), motor programming (Luria’s fist-edge-palm test), sensitivity to interference (conflicting instructions), inhibitory control (go-no go task) and environmental autonomy (prehension behaviour). The FAB total percent score (percentage correct) formed the composite score of executive function.

### 2.4. MRI Data Acquisition and Reconstruction

All patients were scanned on a 3T Philips Achieva MRI scanner (Philips Medical Systems) with a standard 8-channel head matrix coil at the HUS Helsinki Medical Imaging Center at HUCH. The MRI protocol comprised high-resolution T1-weighted anatomical images and whole-brain diffusion-weighted imaging (DWI) data (TR = 11,106 ms, TE = 60 ms, acquisition matrix = 112 × 112, 70 axial slices, voxel size = 2.0 × 2.0 × 2.0 mm^3^) with one non-diffusion weighted volume and 32 diffusion weighted volumes (b = 1000 s/mm^2^).

The DWI data were reconstructed in the Montreal Neurological Institute (MNI) space using q-space diffeomorphic reconstruction (QSDR) [35] that allows the construction of spin distribution functions (SDFs) [36]. The b-table was checked by an automatic quality control routine to ensure its accuracy [37]. Normalization was carried out using the anisotropy map of each participant and a diffusion sampling length ratio of 1.25 was used. The data output was resampled to 2 mm isotropic resolution. Quality of the normalization was inspected using the R^2^ values denoting goodness-of-fit between the participant’s anisotropy map and template as well as inspecting the anatomical localisation of each participant’s forceps major and minor to confirm the normalization quality [29]. The restricted diffusion was quantified using restricted diffusion imaging [38] and QA was extracted as the local connectome fingerprint [39] and used in the connectometry analysis. QA-based tractography has been shown to outperform traditional fractional anisotropy-based methods by being more specific to individual’s connectivity patterns [39] and less susceptible to the partial volume effect of crossing fibres and free water, as well as to provide better resolution in tractography [40].

### 2.5. Regions of Interest

To focus the analyses on the neural structures related to music therapy induced changes in TBI, the Automated Anatomical Labelling atlas 3 (AAL3) [41] was used to define the ROIs based on our previous studies [26,27]. Four regions were derived from the AAL3: right inferior frontal gyrus (pars operculum and triangularis), Rolandic operculum, and inferior parietal lobule.

### 2.6. Data Analysis

Diffusion MRI connectometry [28] analyses were carried out using DSI Studio (http://dsi-studio.labsolver.org, version 7 April 2021). Connectometry was used to derive the correlational tractography that has longitudinal QA changes correlated with Group. To do this, two nonparametric multiple regression models were used to identify local connectome (i.e., QA) changes across time (TP2 > TP1 and TP3 > TP2) between the groups (AB and BA). Local connectomes exceeding a *t*-statistic threshold of 2 were selected and tracked using a deterministic fibre tracking algorithm [40] to obtain correlational tractography. The tracks were filtered by topology-informed pruning [42] with 4 iterations, and a length threshold of 20 voxel distance was used to identify significant tracts. Bootstrap resampling with 10,000 randomized permutations was used to obtain the null distribution of the track length and estimate the false discovery rates (FDR).

To evaluate whether the intervention-induced longitudinal QA changes were associated with behavioural gains in executive function, the mean QA change in the network of significant connectometry results was extracted for each patient and exported to SPSS (IBM SPSS Statistics for Windows, v.27.0. IBM Corp.: Armonk, NY, USA). Then, nonparametric correlations (Spearman, two-tailed) were calculated over the whole sample between the longitudinal mean QA change in the significant connectometry results and the longitudinal FAB score change to determine the structural relationship with behavioural gains. To control for multiple comparisons, FDR-correction was applied.

## 3. Results

The demographic, clinical, and musical background information of the patients is presented in Table 1. There were no significant differences between the AB (*n* = 16) and BA (*n* = 9) groups.

The connectometry analyses comparing the QA changes from baseline (TP1) to 3-month (TP2) stage between the groups revealed that the group receiving the NMT (AB) showed greater QA increase (TP2 > TP1) compared to the group receiving only standard care (BA) in the right dorsal pathways (arcuate fasciculus, superior longitudinal fasciculus, frontal aslant tract) and in the right thalamic radiation and corticostriatal tract (FDR = 0.005; d = 0.97, Figure 2A). The mean increased QA in this network of results correlated with increased FAB score (r = 0.46, *p* = 0.021). No significant QA increases from TP1 to TP2 were observed in the BA group compared to the AB group.

Similar findings were observed when comparing the groups between from the 3-month (TP2) to the 6-month (TP3) stage: the group receiving the NMT (BA) showed greater QA increase (TP3 > TP2) in the right dorsal pathways (arcuate fasciculus, superior longitudinal fasciculus, frontal aslant tract) and in the right corticostriatal tract as well as in the corpus callosum (FDR = 0.009; d = 0.65, Figure 2B) compared to the group receiving only standard care (AB). Significant correlations between the mean increased QA and increased FAB score were not observed. There were no significant QA increases from TP2 to TP3 in the AB group compared to the BA group.

## 4. Discussion

This structural white matter connectometry study set out to determine structural connectome changes induced by NMT and their relation to improved executive function in patients with TBI. Our novel findings were that compared to standard care, NMT enhanced structural connectivity in right frontal dorsal and projection pathways as well as in the corpus callosum. The enhanced structural connectivity was associated with improved executive function. To our best knowledge, this is the first study on patients to link treatment-induced white matter neuroplasticity changes with improved cognitive outcome in TBI. This study provides novel and crucial information about the neural mechanisms of music therapy induced brain recovery following TBI.

Executive dysfunction is considered to be the core clinical feature of TBI [20], particularly in moderate-to-severe cases, on which our study focused on, contributing significantly to both acute and chronic disability. Executive function is a broad term referring to higher order cognitive processes that enable individuals to regulate their thoughts and actions during goal-directed behaviour [43]. These processes are widely thought to include inhibition of prepotent responses, attentional control, task switching and working memory updating [44]. Neurally, executive function contributes to the coordination of these processes across a network of brain structures that need to work in concert. A large-scale quantitative meta-analysis of 193 functional neuroimaging studies has revealed that such executive control network comprises dorsolateral prefrontal, anterior cingulate, and parietal cortices [45]. To allow efficient communication, these spatially distributed brain regions are structurally connected via white matter pathways [46]. The three dominant structural components of this executive control network are (i) interhemispheric connections via the corpus callosum, (ii) fronto-parietal pathways, mainly the superior longitudinal fasciculus connecting dorsolateral prefrontal cortex and the parietal lobe, and (iii) the corticostriatal pathways between right dorsolateral prefrontal cortex and striatum [46,47,48].

The dependence on multiple white matter pathways working in concert makes executive function vulnerable to reduced communication efficiency following DAI in TBI. This was reflected in a recent meta-analysis evaluating the relationships between white matter fractional anisotropy values and executive dysfunction following TBI [49]. Executive dysfunction in TBI was associated with damage to various white matter pathways, but was most significantly correlated with decreased fractional anisotropy values in the superior longitudinal fasciculus, an association tract connecting the frontal, parietal and temporal lobes [49]. Together with commissural tracts, association tracts are most critical for cognition because they transfer information between lobes and hemispheres [50]. They are also most vulnerable to injury in TBI [12,13,14] and undergo long-term degeneration after the initial injury [15,16], giving rise to the functional and cognitive outcomes in TBI [16,17].

The white matter is a biologically active component of the brain, amenable to treatment-induced modifications that bring about beneficial behavioural change [24,51]. In TBI, two principles of white matter structure and function are pertinent to acknowledge in the rehabilitation. First, the severity and prognosis are related to the degree of initial axonal injury, that is, if the axonal scaffolding on which myelin can regain its integrity is preserved, the axonal recovery is plausible, and therefore the restoration of function [52]. Second, white matter structure possesses the capacity of plasticity, that is, modification of its structure by experience [53]. Since neurogenesis has no known clinically meaningful effect on (adult) brain recovery, restoration of function relies upon the ability of spared neurons to grow neurites and form new synapses to rebuild and remodel the injured networks [54,55]. This requires neuronal activity, that is, stimulation via rehabilitation [56] that provides a fertile ground for neuroplasticity and has been shown to produce plastic changes in the white matter including activity-dependent myelination [57] and enhanced oligodendrogenesis [58].

Our present results showing that NMT-induced white matter plasticity changes underpinning the improved executive function comprise the right (i) corpus callosum, (ii) frontoparietal connections, and (iii) the right corticostriatal tract parallel strongly with the previous evidence on the structural core of the executive control network [46,47,48,59] as well as with the evidence on critical white matter damage loci in TBI giving rise to executive dysfunction [49]. Our results are also well in line with previous DTI studies which have reported neuroplasticity changes in the arcuate fasciculus [60,61], superior longitudinal fasciculus [62,63], corpus callosum [63,64], and corticospinal tract [65] induced by musical training in healthy subjects and in the frontal aslant tract induced by music-based rehabilitation in stroke patients [24]. While previous evidence on treatment-related white matter plasticity in TBI is scarce, our results argue that structural connectivity can be enhanced via treatments in TBI and that the treatment-related white matter changes are associated with improved cognitive outcome.

The present study has some potential limitations that need to be considered when evaluating the findings. Due to the study design, it is impossible to infer whether some components of the intervention, for example active or passive engagement during music-making or movements while playing musical instruments, were more relevant to clinical improvement than others. Moreover, although the study is the largest RCT utilizing NMT in moderate-to-severe TBI to date, the sample size remains relatively modest (*n* = 25) and may preclude the detection of smaller neuroplasticity effect sizes due to lack of statistical power. The limited sample size also prevents us from making more fine-grained analyses of what are the differences and commonalities in the patterns of white matter changes and executive function recovery between different time points. Therefore, future longitudinal studies with a stratification of patients by time since injury as well as lesion site would be warranted to explicitly test the relationship of music therapy induced cognitive and neuroplasticity changes across the recovery spectrum.

## 5. Conclusions

In conclusion, the present results suggest that the positive effects of neurological music therapy on executive function recovery in TBI are underpinned by structural white matter reorganization within the structural executive function network. Clinically, together with our previous results [26,27,66], this evidence suggests that music therapy is a feasible tool to induce structural white matter neuroplasticity in the post-TBI brain that underpin improved executive function.

## Figures and Tables

**Figure 1 jcm-11-02184-f001:**
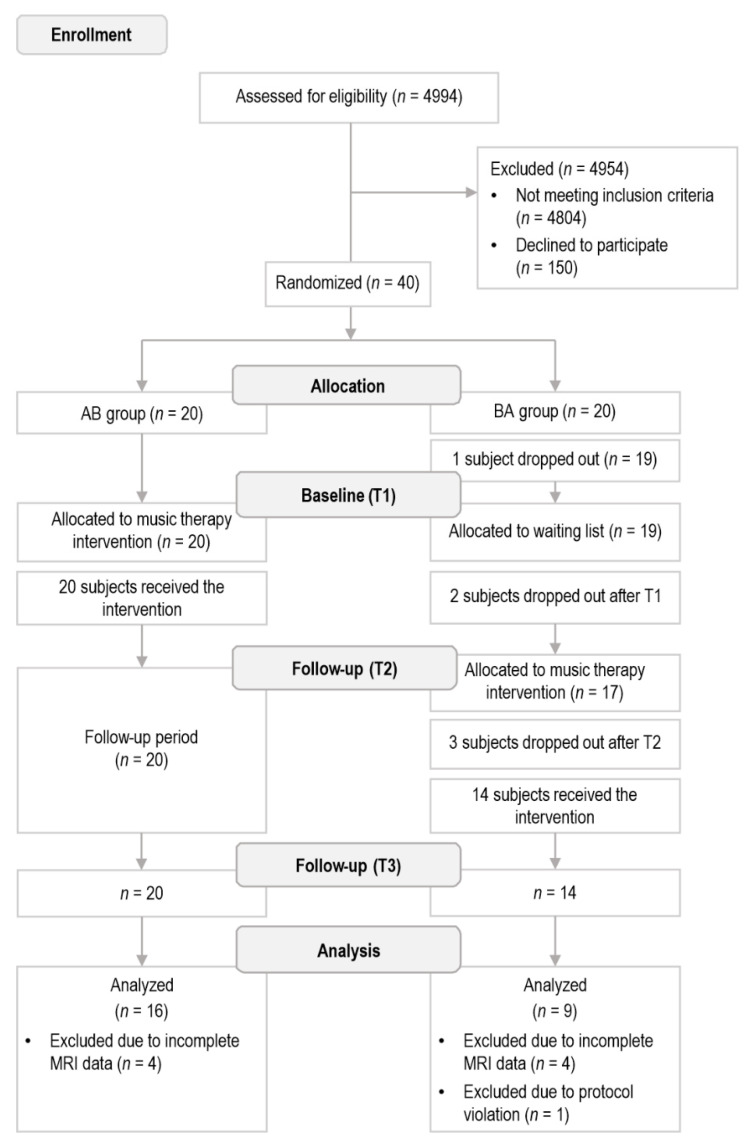
Flow diagram outlining the trial.

**Figure 2 jcm-11-02184-f002:**
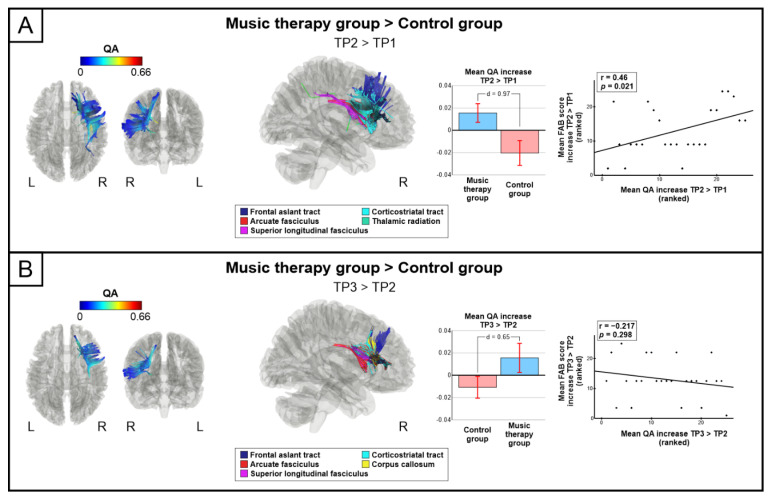
Music therapy induced structural white matter connectometry changes. Significant connectometry changes showing increased structural white matter connectivity between (**A**) music therapy and control group (TP2 > TP1) and (**B**) music therapy and control group (TP3 > TP2). Mean longitudinal QA change correlations (Spearman, two-tailed) to FAB score change are shown with scatter plots. Bar plots for mean QA in the significant connectivity results for both groups are shown: bar = mean, error-bar = standard error of mean, d = Cohen’s d, L = left, QA = quantitative anisotropy, R = Right, TP = time point.

**Table 1 jcm-11-02184-t001:** Demographic, clinical, and musical background information (*n* = 25).

	AB	BA	Difference between Groups (*p*)
**Demographic information**			
Age	42.1 (14.9)	40.8 (11.5)	0.814 (*t*)
Gender (female/male)	7/9	3/6	0.691 (*X*^2^)
Education in years	14.3 (2.7)	14.9 (2.1)	0.535 (*t*)
**Clinical information**			
TBI severity (moderate/severe)	13/3	5/4	0.170 (*X*^2^)
GCS (severe/moderate/minor) ^a^	2/3/10	2/0/6	0.357 (*X*^2^)
PTA classification (mild/moderate/severe) ^b^	9/3/2	3/3/2	0.330 (*X*^2^)
Cause of injury (traffic-related/fall/other)	6/9/1	2/3/4	0.072 (*X*^2^)
Time since injury (months)	8.4 (6.0)	7.1 (6.1)	0.622 (*t*)
Lesion laterality ^c^ (left/right/both)	3/1/11	3/0/6	0.603 (*t*)
DAI ^c^ (yes/no)	6/9	6/3	0.400 (*X*^2^)
Hemorrhages, bleeds or ischemic injury ^c^ (yes/no)	10/5	5/4	0.678 (*X*^2^)
GOSE ^d^	5.2 (1.5)	5.6 (1.1)	0.541 (*t*)
NOS-TBI ^e^	2.0 (2.1)	1.8 (2.3)	0.812 (*t*)
**Musical background**			
Instrument playing (yes/no)	11/6	5/3	1.000 (*X*^2^)
Years of playing	5.2 (11.3)	3.8 (5.8)	0.783 (*t*)
Singing (yes/no)	9/7	3/6	0.411 (*X*^2^)
Years of singing	7.9 (13.7)	1.3 (3.0)	0.170 (*t*)
Dancing (yes/no)	9/7	4/5	0.688 (*X*^2^)
Years of dancing	5.4 (10.9)	4.8 (9.8)	0.889 (*t*)

^a^ 3–8 = severe, 9–12 = moderate, 13–15 = minor; ^b^ 1 = mild (<1 day), 2 = moderate (1–7 days), 3 = severe (>1 weeks); ^c^ Based on MRI findings; ^d^ Glasgow Outcome Scale Extended; ^e^ Neurological Outcome Scale for TBI.

## Data Availability

Anonymized data reported in this manuscript are available from the corresponding author upon reasonable request and subject to approval by the appropriate regulatory committees and officials. No custom codes were used in any of the analyses.

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
