# Peer review of "Neurological Music Therapy Rebuilds Structural Connectome after Traumatic Brain Injury: Secondary Analysis from a Randomized Controlled Trial"

_jcm, 2022, doi:10.3390/jcm11082184_

Round 1

Reviewer 1 Report

jcm-1614155

The manuscript seems to handle a re-analysis of a previous clinical trial already reported as Siponkoski et al. J Neurotrauma 2020, 37, 618–634, doi:10.1089/neu.2019.6413 (Ref #26). The main findings were that neurological music therapy (NMT) increased quantitative anisotropy (QA) of selected tracts in the brain, and that the QA increase was correlated with the executive function measured by FAB in some (i.e. AB) group but not in the other, BA group in patients with traumatic brain injury (TBI). I appreciate the effect of this new therapy, but some issues need clarification and discussion.

1) The fact that this is a re-analysis of a previous clinical trial is not very clear. It should be explicitly mentioned, for example at the beginning of the “Subjects and study design” section, that this is a sort of post hoc analysis of the Ref 26.

2) Although I agree it is inevitable, the study design is single-blinded, namely the patients themselves should know that they received NMT or not. This point should be described.

3) With this study design, it is impossible to infer whether some components, e.g., actual movements of playing instruments or some conception of it, were more relevant to clinical improvement than others. This issue should be clearly argued as a limitation of the study.

4) Regarding the number of patients, 40 were recruited, 6 dropped out, and in 8 MRI was not possible (lines 132 to 139). Then, the total number of patients analyzed should be 40 - 6 - 8 = 26, but the manuscript reads it was 25 (line 138). Please explain.

5) In Table 1, “cause of injury (traffic-related/fall/other)” seems lacking.

6) In Figure 1, please add labels on the X- and Y-axis. Also, even with insignificance, a scatter plot should be shown on Fig1B.

7) There seems some grammatical errors and typos, for example, in Line 59, “showed” should be “shown.”

Author Response

Reviewer #1

The manuscript seems to handle a re-analysis of a previous clinical trial already reported as Siponkoski et al. J Neurotrauma 2020, 37, 618–634, doi:10.1089/neu.2019.6413 (Ref #26). The main findings were that neurological music therapy (NMT) increased quantitative anisotropy (QA) of selected tracts in the brain, and that the QA increase was correlated with the executive function measured by FAB in some (i.e. AB) group but not in the other, BA group in patients with traumatic brain injury (TBI). I appreciate the effect of this new therapy, but some issues need clarification and discussion.

Comment 1: The fact that this is a re-analysis of a previous clinical trial is not very clear. It should be explicitly mentioned, for example at the beginning of the “Subjects and study design” section, that this is a sort of post hoc analysis of the Ref 26.

Reply 1: We thank the Reviewer for the feedback. Change in structural white matter neuroplasticity was a pre-specified secondary outcome of the study (as registered in clinicaltrials.gov, see https://www.clinicaltrials.gov/ct2/show/NCT01956136) and therefore the current analyses are not post hoc in nature. However, we agree with the Reviewer that this could be more clearly expressed in the manuscript and therefore we have revised the title to “Neurological music therapy rebuilds structural connectome after traumatic brain injury: Secondary analysis from a randomized controlled trial”.

Moreover, we revised the Abstract in accordance with CONSORT guidelines (page 2):

Background: Traumatic brain injury (TBI) is a common and devastating neurological condition, associated often with poor functional outcome and deficits in executive function. Due to the neuropathology of TBI, neuroimaging plays a crucial role in its assessment, and while diffusion MRI has been proposed as a sensitive biomarker, longitudinal studies evaluating treatment-related diffusion MRI changes are scarce. Recent evidence suggests that neurological music therapy can improve executive functions in patients with TBI and that these effects are underpinned by neuroplasticity changes in the brain. However, studies evaluating music therapy induced structural connectome changes in patients with TBI are lacking.

Design: Single-blind crossover (AB/BA) randomized controlled trial.

Objective: Here, we report secondary outcomes of the trial and set out to assess the effect of neurological music therapy on structural white matter connectome changes and their association with improved execute function in patients with TBI.

Methods: Using an AB/BA design, 25 patients with moderate or severe TBI were randomized to receive a 3-month neurological music therapy intervention either during the first (AB, n = 16) or second (BA, n = 9) half of a 6-month follow-up period. Neuropsychological testing and diffusion MRI scans were performed at baseline and at the 3-month and 6-month stage.

Findings: Compared to the control group, the music therapy group increased quantitative anisotropy (QA) in the right dorsal pathways (arcuate fasciculus, superior longitudinal fasciculus) and in the corpus callosum and the right frontal aslant tract, thalamic radiation and corticostriatal tracts. The mean increased QA in this network of results correlated with improved executive function.

Conclusion: This study shows that music therapy can induce structural white matter neuroplasticity in the post-TBI brain that underpins improved executive function.

Trial registration: clinicaltrials.gov Identifier NCT01956136”

Comment 2: Although I agree it is inevitable, the study design is single-blinded, namely the patients themselves should know that they received NMT or not. This point should be described.

Reply 2: We have now revised the Abstract (see above or page 2) and Materials and Methods (page 5): “The study was a single-blind crossover RCT with a 6-month follow-up period.” and “All assessments were carried out by research personnel blinded to the patients’ group allocation. Due to the nature of the intervention, patients were not blinded.”

Comment 3: With this study design, it is impossible to infer whether some components, e.g., actual movements of playing instruments or some conception of it, were more relevant to clinical improvement than others. This issue should be clearly argued as a limitation of the study.

Reply 3: Previous evidence (e.g., Tong et al. 2015 Neurol Res) suggests that the motor benefits of music-supported therapy are linked to music itself rather than to repeated practice: patients playing an audible musical instrument demonstrated greater upper limb motor improvement than those playing a mute musical instrument. These beneficial effects are thought to rely on the music-based interventions’ capacity to integrate multimodal stimulation to, for example, motor training, and provide a fertile ground for neuroplasticity (e.g., Grau-Sánchez et al. 2020 Neurosci Biobehav Rev). However, it remains true that it is impossible to tease apart the effects of individual components of the current intervention and we have now included this as a limitation of the study. Please see page 13: “The present study has some potential limitations that need to be considered when evaluating the findings. Due to the study design, it is impossible to infer whether some components of the intervention, for example active or passive engagement during music-making or movements while playing musical instruments, were more relevant to clinical improvement than others.”

Comment 4: Regarding the number of patients, 40 were recruited, 6 dropped out, and in 8 MRI was not possible (lines 132 to 139). Then, the total number of patients analyzed should be 40 - 6 - 8 = 26, but the manuscript reads it was 25 (line 138). Please explain.

Reply 4: We thank the Reviewer for pointing out this unfortunate mistake. One additional patient was excluded from the analysis due to intensive self-implemented piano training during the trial, but this information was mistakenly left out of the manuscript. This information has now been provided in the manuscript and the paragraph revised accordingly (pages 5-6): “Out of the 40 randomized patients, 1 participant dropped out before the TP1 measurements, 2 participants dropped out before TP2, and another 3 participants dropped out before TP3. The dropouts were mainly due to lack of energy and motivation. All dropouts (n = 6) occurred in the BA group, which was likely linked to the long waiting period before the intervention. Of these, five took place before the onset of the intervention. Of the remaining 34 patients, 1 was excluded from the analyses due to intensive self-implemented piano training, which was not part of the trial protocol, and 8 were excluded from the analyses due to lack of MRI data owing to contraindications or technical difficulties during the scanning. Finally, 25 patients (AB: n = 16, BA: n = 9) completed the MRI acquisition in the three time points and were included in the present study.” Please also see the flow chart presented in the Figure 1.

Comment 5: In Table 1, “cause of injury (traffic-related/fall/other)” seems lacking.

Reply 5: We thank the Reviewer for pointing this out. This information has now been included in Table 1.

Comment 6: In Figure 1, please add labels on the X- and Y-axis. Also, even with insignificance, a scatter plot should be shown on Fig1B.

Reply 6: We have now added labels to the X- and Y-axis as well as the scatter plot for Fig 1B. The connectometry analysis in DSI Studio utilizes nonparametric multiple regression model(s) to identify local connectome changes, and to follow this approach, Spearman correlations were carried out between mean QA change in the significant connectometry results and FAB score change over the whole sample to determine the structural relationship with behavioural gains. Unfortunately, the correlation was mistakenly named as Pearson correlation in the previous version of the manuscript, but has now been corrected to Spearman, please see page 8 and Fig 1.

Comment 7: There seems some grammatical errors and typos, for example, in Line 59, “showed” should be “shown.”

Reply 7: We thank the Reviewer for pointing this out. We have corrected this and carefully proofread the manuscript and made corrections when needed.

Reviewer 2 Report

This was an interesting and innovative study assessing the effect of NMT on neuroplasticity after TBI. I encourage the authors to consider the following comments and recommendations. 

Was your trial registered prior to its conduction (if not, please add limitation)?

Is there a formal study protocol available online (if not, please add limitation)?

Please report your study using the CONSORT guidelines.

Please be more specific with the eligibility criteria of the participants. Did you apply any criterion regarding the duration of loss of consciousness (e.g., > 30 min)? Were cases with cognitive impairment in domains other than executive function, attention and memory excluded?

When randomization is performed in batches of two, then allocation concealment is compromised (knowing the lesion laterality, one could recognize the allocation trend quite easily). Please provide information about who enrolled participants and who assigned participants to interventions. Was he/she the same person? Was he/she the same person that generated the random allocation sequence?

How did you address potential practice effects due to repeated assessments?

Please briefly define standard care.

As I understand, participants were recruited months after the TBI event. This makes your design rather vague. Have you used record linkage (ICD-10) to find potential candidates? If so, lease do not mix eligibility criteria with recruitment strategies. Please describe in detail the data selection process (patient files?). Then describe the process of extending an invitation to eligible participants (via telephone?). Then provide a detailed patient flow-chart, indicating the number of screened records, the numbers of eligible and ineligible participants (with reasons), the numbers of individuals that accepted and declined your invitation (with reasons), and the numbers of those lost to follow-up (provide specific details about the exact time of dropout and reasons). Finally, add the important limitation of retrospective data collection (selection bias cannot be ruled out).

Table 1: You should present the baseline characteristics according to your inclusion criteria (GCS and PTA should be treated as categorical and not scale variables). How many patients were included per group based on these criteria (define moderate and severe TBI in the methods according to the duration of the amnesia and the GCS score, then provide absolute frequencies per category in Table 1)? Also, GOSE and NOS-TBI are first mentioned in Table 1 (provide a relevant description in the methods). No numbers are provided for the ‘‘cause of injury’’ variable.  Finally, table 1 belongs to the results section.

Two multiple regression models were used to identify positive local connectome changes across time (TP2 > TP1 and TP3 > TP2) between the AB and BA. This is a deficient description. Please be more specific (what type of regression? What do you mean by positive connectome changes over time? Which were the dependent and independent variables? Did you account for intercorrelations -within the same person measurements are highly correlated-? Shortly after, please define T-score)

Correlations were sought between mean QA change in the significant connectometry results and FAB score change over the whole sample. This part is incomprehensible (paired observations are required to investigate potential correlations, please be specific: which variables were analysed and which participant set was utilised in each analysis).

Please add a table with the crude numbers of the analysed variables over the three time points per group (both neuropsychological and neuroimaging parameters).

Have you considered looking into the different subtests of FAB in order to identify more specific correlations? What about memory or attention (which were mentioned in the eligibility criteria)?

In the limitations section, there is a serious mistake. Small sample sizes limit investigational power, which is translated into smaller precision, i.e., larger confidence intervals. Effect sizes remain unaltered (methodological flaws may alter an effect size). Either smaller or larger, effect sizes should not be confused with precision estimates.

Author Response

Reviewer #2

This was an interesting and innovative study assessing the effect of NMT on neuroplasticity after TBI. I encourage the authors to consider the following comments and recommendations.

Comment 1: Was your trial registered prior to its conduction (if not, please add limitation)? Is there a formal study protocol available online (if not, please add limitation)? Please report your study using the CONSORT guidelines.

Reply 1: We thank the Reviewer for the feedback. Yes, the trial was registered prior to its implementation in Clinical Trials (NCT01956136; https://clinicaltrials.gov/ct2/show/NCT01956136). Detailed descriptions as well as outcome measures are included in the trial registration. However, a formal study protocol paper was not published. We have adhered to the CONSORT guidelines in reporting the study (a CONSORT checklist and flow chart is now enclosed).

Comment 2: Please be more specific with the eligibility criteria of the participants. Did you apply any criterion regarding the duration of loss of consciousness (e.g., > 30 min)? Were cases with cognitive impairment in domains other than executive function, attention and memory excluded?

Reply 2: We acknowledge that our description of the eligibility of the patients has not been complete and clear enough in the manuscript. All patients had to have moderate or severe TBI, diagnosed according to the International Statistical Classification of Diseases and Related Health Problems, 10th revision (ICD-10), fulfilling the criteria of at least moderate severity (Glasgow Coma Scale [GCS] score: ≤12 and/or loss of consciousness >30 min and/or post-traumatic amnesia [PTA] ≥24 h and/or positive findings on CT/MRI), following the formal TBI Current Care guidelines used in Finland. Regarding the cognitive impairment criterion, we included those patients who had deficits in executive function, attention, or memory because our music intervention targeted these cognitive functions. We have now revised the Subjects and study design, please see page 4.

Comment 3: When randomization is performed in batches of two, then allocation concealment is compromised (knowing the lesion laterality, one could recognize the allocation trend quite easily). Please provide information about who enrolled participants and who assigned participants to interventions. Was he/she the same person? Was he/she the same person that generated the random allocation sequence?

Reply 3: When randomizing the patients, we had three separate lists for lesion laterality (left / right / bilateral) within which the patients were randomized in batches of two consecutive patients. This means that, for example, when two patients with a left hemisphere lesion were successfully recruited, they were then randomized to AB/BA. The randomization list was created by the PI of the study (Teppo Särkämö) who was not otherwise involved in the implementation of the data collection. The enrolment was performed by a research nurse (Vera Lotvonen) who did not have access to the randomization list. Only the music therapists (Sari Laitinen, Milla Holma, Mirja Ahlfors) had access the randomization list and they allocated the patients to the AB and BA groups. Therefore, allocation concealment was not compromised in the study.

Comment 4: How did you address potential practice effects due to repeated assessments? Please briefly define standard care.

Reply 4: In order to minimize potential practice effects due to repeated assessments, we used parallel versions those cognitive tests which had a memory component (i.e., specific items to learn/recall) in TP1, TP2 and TP3. Standard care comprised any physical therapy, occupational therapy, speech therapy or neuropsychological rehabilitation which the patients received in public (or private) healthcare during the study period. There were no statistically significant differences between the AB and BA groups in the amount of received standard care; this has already been reported for the current sample in the Supplementary Table S2 of our previous paper (Siponkoski et al. 2020 J Neurotrauma), see https://www.liebertpub.com/doi/suppl/10.1089/neu.2019.6413/suppl_file/Supp_Table2.pdf.

We have also included this information now in the manuscript, please see page 5.

Comment 5: As I understand, participants were recruited months after the TBI event. This makes your design rather vague. Have you used record linkage (ICD-10) to find potential candidates? If so, lease do not mix eligibility criteria with recruitment strategies. Please describe in detail the data selection process (patient files?). Then describe the process of extending an invitation to eligible participants (via telephone?). Then provide a detailed patient flow-chart, indicating the number of screened records, the numbers of eligible and ineligible participants (with reasons), the numbers of individuals that accepted and declined your invitation (with reasons), and the numbers of those lost to follow-up (provide specific details about the exact time of dropout and reasons). Finally, add the important limitation of retrospective data collection (selection bias cannot be ruled out).

Reply 5: The patients were recruited from three centres (Brain Injury Clinic of the Helsinki University Central Hospital, Validia Rehabilitation Helsinki, and the Department of Neurology of the Lohja Hospital). In each centre, we used medical records (patient files) to identify those patients who had been treated for TBI within the previous 24 months and were then assessed for their eligibility (total n=4994). Of these, 4808 did not meet our inclusion criteria and 190 were contacted via telephone and invited to participate. Of these, 150 declined to participate and 40 agreed and were then randomized and allocated to AB/BA. A flow chart documenting this information and the number of patients lost to follow-up was reported in our previous paper for the whole sample (Siponkoski et al. 2020 J Neurotrauma), but for the sake of clarity we have now included a new flow chart for the current sample (n=25). We are not sure what Reviewer #2 means by “retrospective data collection” as all the outcome measure data used in the present study was collected prospectively.

Comment 6: Table 1: You should present the baseline characteristics according to your inclusion criteria (GCS and PTA should be treated as categorical and not scale variables). How many patients were included per group based on these criteria (define moderate and severe TBI in the methods according to the duration of the amnesia and the GCS score, then provide absolute frequencies per category in Table 1)? Also, GOSE and NOS-TBI are first mentioned in Table 1 (provide a relevant description in the methods). No numbers are provided for the ‘‘cause of injury’’ variable.  Finally, table 1 belongs to the results section.

Reply 6: We thank the Reviewer for these detailed comments. GCS and PTA have now been treated as categorical variables and Table 1 has been revised accordingly. Please also see response to Comment 2: AB group had 12 patients with moderate TBI and 3 with severe TBI whereas BA group had 5 patients with moderate TBI and 4 patients with severe TBI [X2(1, N=25)=1.886, p=0.170]. Moderate and severe TBI have now been defined in the Methods, please see page 5. Moreover, we have included a brief description of GOSE and NOS-TBI in the Methods, please see page 5: “Both the extended Glasgow Outcome Scale (GOSE) [31] and the Neurological Outcome Scale for Traumatic Brain Injury (NOS-TBI) [32] were administered to obtain information of the overall symptoms and current functional outcome after TBI.” Information regarding the cause of injury has now been included in Table 1. In addition, Table 1 has been relocated to the Results.

Comment 7: Two multiple regression models were used to identify positive local connectome changes across time (TP2 > TP1 and TP3 > TP2) between the AB and BA. This is a deficient description. Please be more specific (what type of regression? What do you mean by positive connectome changes over time? Which were the dependent and independent variables? Did you account for intercorrelations -within the same person measurements are highly correlated-? Shortly after, please define T-score).

Correlations were sought between mean QA change in the significant connectometry results and FAB score change over the whole sample. This part is incomprehensible (paired observations are required to investigate potential correlations, please be specific: which variables were analysed and which participant set was utilised in each analysis).

Reply 7: Group connectometry analysis utilizes QA-based tractography and allows identifying the exact segment/subcomponent/branches of white matter tracks that are correlated to group differences. As described in the MRI data acquisition and reconstruction, the data are reconstructed and QA extracted as the local connectome fingerprint (i.e., local connectome) to be used in the connectometry analysis. The data analysis pipeline tracts the precise segment of pathways that correlate with the group difference (here, the positive connectome change) and then uses permutation testing for statistical inference, deriving the false discovery rate of the findings. It utilizes nonparametric Spearman rank-based correlation with no normality assumption.

If we understand correctly what Reviewer #2 is referring to with “accounting for intercorrelations within the same person”, the longitudinal local connectome fingerprints are calculated for focal differences between the time points and as only one “predictor”, that is, the Group, is being used in the analysis, intercorrelation or multicollinearity should not be an issue. The t-score is equivalent to t-value. To avoid confusion, we have now revised the paragraph to be more precise, please see pages 8-9:

“Diffusion MRI connectometry [28] analyses were carried out using DSI Studio (http://dsi-studio.labsolver.org, version April 7 2021). Connectometry was used to derive the correlational tractography that has longitudinal QA changes correlated with Group. To do this, two nonparametric multiple regression models were used to identify local connectome (i.e., QA) changes across time (TP2 > TP1 and TP3 > TP2) between the groups (AB and BA). Local connectomes exceeding a t-statistic threshold of 2 were selected and tracked using a deterministic fiber tracking algorithm [39] to obtain correlational tractography. The tracks were filtered by topology-informed pruning [41] with 4 iterations, and a length threshold of 20 voxel distance was used to identify significant tracts. Bootstrap resampling with 10,000 randomized permutations was used to obtain the null distribution of the track length and estimate the false discovery rates (FDR).

To evaluate whether the intervention-induced longitudinal QA changes were associated with behavioural gains in executive function, the mean QA change in the network of significant connectometry results was extracted for each patient and exported to SPSS (IBM SPSS Statistics for Windows, v.27.0. Armonk, NY: IBM Corp.). Then, nonparametric correlations (Spearman, two-tailed) were calculated over the whole sample between the longitudinal mean QA change in the significant connectometry results and the longitudinal FAB score change to determine the structural relationship with behavioural gains. To control for multiple comparisons, FDR-correction was applied.”

Comment 8: Please add a table with the crude numbers of the analysed variables over the three time points per group (both neuropsychological and neuroimaging parameters).

Reply 8: Regarding the neuroimaging (QA) parameters, we included local connectomes exceeding a t-statistic threshold of 2, which were then tracked using a deterministic fiber tracking algorithm, corrected for the number of comparisons using false discovery rates (FDR). Connectomes were restricted to four ROIs (right pars operculum and triangularis, right Rolandic operculum, and right inferior parietal lobule) based on our previous studies (Siponkoski et al. 2020 J Neurotrauma; Martínez-Molina et al. 2021 Neural Plast). Only QA-based analyses were conducted and no other diffusion metrics were analyzed. FA-based analyses could be carried out using similar methodology, but these have been shown to be less specific to individual's connectivity patterns (Yeh et al. 2016 PLoS Comput. Biol) and more susceptible to the partial volume effect of crossing fibers and free water as well as to provide poorer resolution in tractography (Yeh et al. 2013 PLoS One).

Regarding the neuropsychological parameters, we included only one test of executive function (FAB total score, in percentage correct), which we showed in our previous paper (Siponkoski et al. 2020 J Neurotrauma) to improve after the music intervention, also in the current smaller (n=25) sample. The correlation analyses between the longitudinal FAB score and the longitudinal mean QA changes in the significant connectometry results were also FDR-corrected. Therefore, we argue that we have effectively controlled for the multiple comparison issue in the present study.

Comment 9: Have you considered looking into the different subtests of FAB in order to identify more specific correlations? What about memory or attention (which were mentioned in the eligibility criteria)?

Reply 9: This is a good question. While different cognitive domains (including attention and memory) were extensively assessed and their results are reported in our previous paper (Siponkoski et al. 2020 J Neurotrauma), the primary outcome in the trial was executive function defined as the FAB total score (percentage correct). In the primary analysis of the trial (Siponkoski et al. 2020 J Neurotrauma), the results showed that general executive function, as indicated by the FAB, improved more in the AB group than in the BA group over the first 3-month period. Due to this, and to maintain uniformity and avoid the multiple comparison issue, only the FAB total score was used in the current manuscript to evaluate the brain-behaviour relationship of the observed treatment-induced structural white matter changes.

Comment 10: In the limitations section, there is a serious mistake. Small sample sizes limit investigational power, which is translated into smaller precision, i.e., larger confidence intervals. Effect sizes remain unaltered (methodological flaws may alter an effect size). Either smaller or larger, effect sizes should not be confused with precision estimates.

Reply 10: We presume that the Reviewer is referring to page 11: “Although the study is the largest RCT utilizing NMT in moderate-to-severe TBI to date, the sample size remains relatively modest (N = 25) and may preclude the detection of smaller neuroplasticity effect sizes due to lack of statistical power.” Here we aimed to discuss how the current limited sample size affects statistical power and that if the effect size of the intervention is large, it is possible to detect such an effect in the present study, whereas detecting smaller but significant effects would require larger sample sizes (e.g., Sullivan and Feinn 2012 J Grad Med Educ).

Round 2

Reviewer 2 Report

Thank you for considering my recommendations